# Mathematical Modelling for Optimal Vaccine Dose Finding: Maximising Efficacy and Minimising Toxicity

**DOI:** 10.3390/vaccines10050756

**Published:** 2022-05-11

**Authors:** John Benest, Sophie Rhodes, Thomas G. Evans, Richard G. White

**Affiliations:** 1Department of Infectious Disease Epidemiology, London School of Hygiene and Tropical Medicine, Keppel Street, London WC1E 7HT, UK; sophie.rhodes@lshtm.ac.uk (S.R.); richard.white@lshtm.ac.uk (R.G.W.); 2Vaccitech Ltd., The Schrodinger Building, Heatley Road, The Oxford Science Park, Oxford OX4 4GE, UK; tom.evans@vaccitech.co.uk

**Keywords:** dosing, dose response, modelling, clinical trials, adaptive design, continual modelling

## Abstract

Vaccination is a key tool to reduce global disease burden. Vaccine dose can affect vaccine efficacy and toxicity. Given the expense of developing vaccines, optimising vaccine dose is essential. Mathematical modelling has been suggested as an approach for optimising vaccine dose by quantitatively establishing the relationships between dose and efficacy/toxicity. In this work, we performed simulation studies to assess the performance of modelling approaches in determining optimal dose. We found that the ability of modelling approaches to determine optimal dose improved with trial size, particularly for studies with at least 30 trial participants, and that, generally, using a peaking or a weighted model-averaging-based dose–efficacy relationship was most effective in finding optimal dose. Most methods of trial dose selection were similarly effective for the purpose of determining optimal dose; however, including modelling to adapt doses during a trial may lead to more trial participants receiving a more optimal dose. Clinical trial dosing around the predicted optimal dose, rather than only at the predicted optimal dose, may improve final dose selection. This work suggests modelling can be used effectively for vaccine dose finding, prompting potential practical applications of these methods in accelerating effective vaccine development and saving lives.

## 1. Introduction

Vaccination is a key tool in global disease burden reduction and disease prevention. However, developing a vaccine for clinical use is an expensive and time-consuming process. As the magnitude of vaccine dose amount (hereafter ‘dose’) can affect the efficacy, toxicity, and cost of administering the vaccine, finding optimal vaccine dose is important. It is important to ensure that the chosen dose best balances maximal efficacy and minimal toxicity [1]. Preclinical and early phase 1/2 dose-finding trials aim to achieve this, typically through direct comparison of the efficacy and toxicity profiles of a small number of doses [2]. However, if none of these small number of doses is the optimal dose, then the vaccine will proceed to further study or clinical use with a suboptimal dose. This could reduce the potential for disease burden reduction, either due to reduced vaccine efficacy or decreased vaccine uptake arising from increased risk of vaccine-related adverse events. Hence, choosing from only a small number of doses may cost lives and be wasteful, given the expense of vaccine development. However, generating data on a larger number of doses requires larger and more expensive trials.

Mathematical-modelling-based approaches for vaccine dose optimisation have been explored previously and represent a solution for identifying optimal dose amongst a large number of possible doses without greatly increasing the size of trials [3,4,5,6]. Under these approaches, rather than comparing the efficacy and toxicity data directly between dosing groups, the data are used to inform models that attempt to describe the dose–efficacy and dose–toxicity relationships. These models are then combined and used to inform vaccine dose decision making, similar to the idea of ‘model-based drug development’, which is prevalent in selecting optimal drug dose [7]. These ‘models’ are equations or systems of equations that are used to describe the relationship between vaccine dose and vaccine response. These models can either be mechanistic, leveraging knowledge of immunodynamics to describe an approximation of the dose-dependent immune system dynamics [3,4], or statistical, using a simpler set of assumptions about the general nature of the relationship between dose and efficacy/toxicity [8,9,10,11]. Although both approaches have been used to explore vaccine dose optimisation, neither approach has been fully validated, accepted, and used. We will be discussing statistical models of dose efficacy and dose toxicity throughout the remainder of this work.

Given that modelling is not consistently used in finding optimal vaccine dose, there are a number of questions that arise concerning its implementation. Firstly, which types of mathematical models would be most useful for determining optimal dose? Secondly, how many individuals in the trial population are required for modelling to generate reliable evidence? Thirdly, how should the trial population be dosed to improve the model’s ability to determine optimal dose? This final question include is whether modelling should be used only retrospectively (as has been done in the past [3,4,11]) or whether it should be used continually at interim timepoints to guide dose selection throughout the trial (in the style of adaptive design or continual reassessment modelling [12,13,14]) in combination with retrospective modelling. Continual modelling/dose recommendation approaches have previously been suggested to be a more ethical approach to conducting dose-ranging studies in drugs [14,15].

Although modelling has previously been applied in vaccine dose optimisation using real-world data, such data are often noisy, and true underlying dose–efficacy and dose–toxicity relationships are unknown. This means that whether the doses that have been selected by these dose-optimisation approaches are truly optimal is unknown. By simulating clinical trial data, where the underlying dose–efficacy and dose–toxicity relationships are known, a ‘simulation study’ [15,16,17] allows for analysis of these dose-optimsation approaches not hampered by noisy data [17,18].

In this work, we aimed to use simulation of dose-finding clinical trials to assess the capability of statistical mathematical models to determine optimal dose. To answer the questions posed above, we investigated modelling-based dose-optimisation approaches, which were defined by:i.Assumed statistical efficacy model.ii.Trial size.iii.Method of trial dose selection.

In order to perform this analysis, we used a number of qualitatively different ‘scenarios’, each representing a different ‘true’ vaccine dose–efficacy and dose–toxicity relationship. We considered three metrics of the quality of a dose-optimsation approach: not only the quality of the final selected dose but also the accuracy of predictions and benefit to trial participants.

Specifically, our objectives were to investigate, through simulation studies over many qualitatively different scenarios:When the method of trial dose selection is fixed, how dose-optimisation approaches are affected by the assumed statistical efficacy model and trial size.When trial size is fixed, how dose-optimisation approaches are affected by the assumed statistical efficacy model and method of trial dose selection.

## 2. Materials and Methods

Here, we summarise the simulation study used in this approach, then detail the component parts.

### 2.1. Overview of Simulation Study Methodology

We would like to evaluate how (i) assumed statistical efficacy model, (ii) trial size, and (iii) method of trial dose selection affect how well optimal dose is determined and use a simulation study to do this (Figure 1). Optimal dose was defined as a function that aims to maximise efficacy and minimise toxicity (3.2). We propose ‘dose-optimsation approaches’, which vary in (i–iii) (3.4). These varying dose-optimsation approaches are tested by simulating clinical trials. These clinical trials are simulated using a number of ‘scenarios’ representing theoretical vaccines that could be optimised (3.3). Each clinical trial was a pairing of a dose-optimsation approach and a scenario and therefore represents how well that dose-optimisation approach could optimise the vaccine represented in that scenario.

The fact that the ‘true’ dose–efficacy and dose–toxicity curves are known in these scenarios allows these dose-optimisation approaches to be assessed. By repeatedly simulating different dose-optimisation approaches/scenarios we can evaluate the effect of varying (i–iii). Using different scenarios reduces the probability that we would recommend a dose-optimisation approach that does not optimise dose well in general, despite optimising dose well in simulations.

### 2.2. Efficacy, Toxicity, and Utility

We introduce the concept of dose-utility as a function of dose efficacy and dose toxicity and define the mathematical models that we used to describe these relationships.

#### 2.2.1. Dose Efficacy

Vaccine efficacy or protection can be defined by many clinical endpoints, for example, reduced risk of infection, reduced risk of symptoms, reduced risk of severe symptoms, or reduced risk of hospitalisation [19]. Without the use of challenge studies or larger phase 3 studies determining relative reduction in disease, the probability of protection can be difficult to determine [20]. Instead, immunological data are typically used in early trials as an anticipated surrogate of protection [21].

Although immunological data may be continuous in nature, a predictive model between immunological readout and probability of efficacy is often unknown [22]. Hence, it is common to define a threshold and consider individuals with immune response in excess of that threshold to have experienced an efficacious response [23]. Therefore, the actual desired endpoint (e.g., protection/survival) is likely binary in nature, and surrogates are often also binary. For simplicity and to aid in general usability, we therefore assumed that for dose-ranging studies, there would exist a binary efficacy outcome that can be measured and that the probability of this binary efficacy outcome is aimed to be maximised.

Even under these assumptions, there was a further challenge in modelling dose-efficacy. Whereas for many drugs, we can assume that an increased dose increases efficacy, for vaccines, this may not be the case. It is possible that there exists some dose for which the probability of efficacy is maximised and that increasing this dose decreases the probability of efficacious response [3,24,25,26]. Below, we define approaches for modelling vaccine efficacy. We chose a sigmoidal curve to represent the former “saturating” dose–response curve shape (Figure 2a) and a latent quadratic function to represent the latter “peaking” dose–response curve shape (Figure 2b). These equations are presented below and have previously been suggested in the literature [27,28]:(1)$$\mathrm{Saturating}\left(\mathrm{Dose}\right)=\frac{\mathrm{maximum}}{1+e^{\left(\mathrm{gradient}\left(\mathrm{midpoint}-\mathrm{Dose}\right)\right)}}$$
(2)$$\mathrm{Peaking}\left(\mathrm{Dose}\right)=\frac{1}{1+e^{\left(\mathrm{base}+\mathrm{gradient}1\times \;\mathrm{Dose}+\mathrm{gradient}2\times {\mathrm{Dose}}^{2}\right)}}$$

Further details of these models can be found in Appendix A.

In the case of uncertainty in the true dose–efficacy shape, a model averaging technique could also be considered [29]. Here, the saturating model and peaking models make predictions and are then combined based on how well each model describes the data. The mathematics behind this are discussed in Appendix A and [22] and a visual depiction is presented in (Figure 3).

#### 2.2.2. Dose Toxicity

As vaccine adverse events are typically less severe than adverse events for drugs and vaccines are preventive rather than therapeutic, we decided that only modelling higher-grade adverse events was unrealistic. It is likely that in vaccine dose-optimisation, minimising lower-grade adverse events may be preferable and relevant to vaccine uptake. Hence, we modelled vaccine dose toxicity using the ordinal dose-toxicity model [15]. Here, toxicity was described by four grades using a four-level toxicity grading system (Table 1).

We modelled the relationship between dose and the ordinal toxicities using the probit method described in [15] and discussed further in Appendix A. A visual description of an example ordinal model is given in Figure 4. Four parameters were needed to define this model. Three parameters defined the dose thresholds for which at least 50% of individuals experience greater than grade 0, grade 1, and grade 2 adverse events. The final parameter defined the steepness of these thresholds.

#### 2.2.3. Dose Utility

Optimising vaccine dose can be considered a multi-objective optimisation problem, in which we aim to maximise efficacy and minimise toxicity. To better define this problem, we made use of a utility function that attempts to balance maximising efficacy and minimising toxicity in a manner that should be clinically meaningful (Appendix A). Although many utility functions might be reasonable, to reduce complexity, a simple and interpretable dose-utility function was chosen [32].

For each dose, we assumed that there is some (predicted or true) probability of efficacy, P(Efficacy|Dose). Additionally, we assume that there are probabilities for each grade of toxicity, P(Toxicity = 0|Dose), P(Toxicity = 1|Dose), P(Toxicity = 2|Dose), and P(Toxicity = 3|Dose). We then defined utility weights, which were:Weight_Efficacy_DisabilityWeight_Toxicity0_DisabilityWeight_Toxicity1_DisabilityWeight_Toxicity2_DisabilityWeight_Toxicity3_

These were measures of how beneficial an efficacious response was relative to the detrimental effect of the different adverse event grades. For example, if Weight_Efficacy_ > DisabilityWeight_Toxicity2_, then the protection that may be gained from an efficacious vaccine response would outweigh the discomfort of the grade 2 event. Conversely, if Weight_Efficacy_ < DisabilityWeight_Toxicity3_, then the protection that may be gained from an efficacious vaccine response would be outweighed by the discomfort of the grade 3 event. The disability weight for each grade was increasing (i.e., a grade 2 adverse event was worse than a grade 1 adverse event) (Table 2).

The dose-utility function is given by:(3)$$\mathrm{Utility}\left(\mathrm{Dose}\right)={\;\mathrm{Weight}}_{\mathrm{Efficacy}}\times P\left(\mathrm{Efficacy}|\mathrm{Dose}\right)-\mathrm{WeightedToxicity}\left(\mathrm{Dose}\right)$$
(4)$$\mathrm{WeightedToxicity}\left(\mathrm{Dose}\right)=\sum_{\mathrm{Grade}=0}^{3}P(\mathrm{Toxicity}=\mathrm{Grade}|\mathrm{Dose})\times {\mathrm{DisabilityWeight}}_{\mathrm{Grade}}$$

A similar idea of vaccine risk/benefit is discussed in relation to the recent COVID-19 AstraZeneca vaccine [33]. Weight_Efficacy_ would vary depending on the disease’s severity, prevalence, and level of confidence in the surrogate of protection. Hence, in this work, we chose Weight_Efficacy_ to be similar relative to DisabilityWeight_Toxicity3_ (Table 2). This ensures that both maximising efficacy and minimising toxicity are important and prevents the optimal dose from being one that is optimal with regards to only one of these goals. Practically, Weight_Efficacy_ could be chosen based on epidemiological models [34].

**Table 2 vaccines-10-00756-t002:** Disability and efficacy weights for the utility functions.

Weight	Value	Source
Weight_Efficacy_	0.133 or 0.266	Chosen to be equal to either DisabilityWeight_Toxicity3_ or twice DisabilityWeight_Toxicity3_
DisabilityWeight_Toxicity0_	0.000	Chosen to be 0, as no discomfort/toxicity is caused
DisabilityWeight_Toxicity1_	0.006	[35]
DisabilityWeight_Toxicity2_	0.051	[35]
DisabilityWeight_Toxicity3_	0.133	[35]

### 2.3. Scenarios

We considered it preferable to ensure that any dose-optimisation approaches that are used in clinical practice are ‘consistent’, which is to say that they optimise dose well for any vaccine they are applied to [36]. The opposite possibility would be for a dose-optimisation approach to be ‘overly specific’, which is to say that the approach would optimise dose very well for a small number of possible vaccines but would fail to choose a good dose for the majority of possible vaccines. To test whether these dose-optimisation approaches were ‘consistent’, we generated a number of qualitatively different ‘scenarios’ that dose-optimsation approaches could be tested on, similar to the study designs used in other dose-optimisation modelling studies [17,28].

Scenarios can be considered as simulated potential ‘truths’ for future vaccine dose/toxicity/response characteristics. Here, a ‘scenario’ was defined by a dose–efficacy curve, a dose–toxicity curve, and utility weights, i.e., the dose–utility curve resulting from these three scenarios (Figure 1, blue box). These scenarios were defined in order to be qualitatively different from each other, covering a broad range of potential dose/toxicity/response characteristics, not based on historical data.

We created and then tested our approaches on 14 such scenarios. For their true dose–efficacy curves, five scenarios used the sigmoid saturating curve, another five scenarios used the latent quadratic peaking curve, and the remaining four scenarios used curves that deviate from the parametric form of those two curves. Visualisations for three of the scenarios are shown in Figure 5, and further visualisation and parameterisation for all 14 scenarios can be found in Appendix A.

### 2.4. Dose-Optimisation Approaches

A dose-optimisation approach can be considered as the combined approach by which a vaccine dose-finding study is conducted, data are gathered, and an ‘optimal’ dose is chosen based on these data. Although there are many possible considerations for doing so, we only considered a subsection of modelling-based dose-optimisation approaches. Therefore (Figure 1, red boxes), for the purposes of this work, a dose-optimisation approach was defined as a combination of:i.An assumed efficacy model (saturating, peaking, or weighted);ii.A trial size (10/30/60/100);iii.A method of trial dose selection (with either retrospective or continual modelling).

Objective 1 focuses on i and ii, and objective 2 focuses on i and iii.

### 2.5. Additional Details

Throughout this work, we considered dose on a log_10_ scale, although we did not otherwise assume units. For viral vector vaccines, these units would likely be viral particles or infectious units. Additionally, we consistently used a dose range of 0–10 on the log_10_ scale. This was purely for convenience and could be rescaled to the minimum and maximum possible dose for any given vaccine. This is referred to as the ‘dosing space’.

For all models, parameter estimation was conducted by minimising negative log likelihood. This was done using the simplex method of Nelder and Mead [37] with the SciPy optimisation package in python [38]. Bounds were placed on parameters to ensure biological plausibility, see Appendix A.

### 2.6. Objective 1: When the Method of Trial Dose Selection Is Fixed, How Dose-Optimisation Approaches Are Affected by the Assumed Statistical Efficacy Model and Trial Size

We first assessed the use of dose-optimisation approaches using the three models of dose efficacy discussed above (saturating, peaking, and weighted) with regards to retrospective modelling with various trial sizes. Using the definition of a dose-optimisation approach outlined above, we assessed the following approaches:i.Efficacy model: saturating, peaking, or weighted;ii.Trial dose-selection method: full uniform exploration;iii.Trial size: 10, 30, 60, or 100.

The method of dose selection for this objective was ‘full uniform exploration’. This method distributes trial participants uniformly over the dosing space. For example, if there were only 6 available trial participants over the [0–10] log_10_-scale dosing space, we would have assigned test doses at 0, 2, 4, 6, 8, and 10. This method of dose selection is reasonable as a naive method, as it would ensure that all areas of the dosing space were evenly explored. As these data would then be a representative sample of all possible doses, this should have allowed for good model calibration and hence a good suggestion of optimal dose.

We assessed 4 different trial sizes explored in this objective. These were 10, 30, 60, and 100 individuals, representing reasonable sizes for vaccine phase I and II trials [39,40,41]. Hence, there were 12 (=4 × 3) dose-optimisation approaches, reflecting a combination of the 4 trials sizes and 3 assumed efficacy models. Each scenario/approach pairing was simulated 100 times for a total of 16800 (=12 × 14 × 100) simulated trials and 840,000 simulated individuals.

#### 2.6.1. Metrics for Comparison between Approaches

We compared dose-optimisation approaches by calculating ‘simple regret’, ‘percentage simple regret’, ‘inaccuracy’, ‘absolute inaccuracy’, ‘average regret’, and ‘percentage average regret’ for each simulation (Figure 6).

##### Simple Regret

Simple regret in this setting was defined by the true utility score of the predicted optimal dose compared to the true optimal utility for the given vaccine. Ideally, this should be minimised. This is shown in Figure 6a and given by the following formula:(5)$$\mathrm{Simple}\;\mathrm{Regret}={\;\mathrm{Utility}}_{\mathrm{TrueOptimal}}-{\;\mathrm{Utility}}_{\mathrm{Chosen}}$$

As the maximum and minimum possible utilities varied between scenarios, we also used the percentage simple regret (PSR) metric to allow for meaningful comparison across combinations of scenarios. PSR is given by the following formula:(6)$$\mathrm{PSR}=100\times \frac{{\mathrm{Utility}}_{\mathrm{TrueOptimal}}-{\;\mathrm{Utility}}_{\mathrm{Chosen}}}{{\mathrm{Utility}}_{\mathrm{TrueOptimal}}-{\;\mathrm{Utility}}_{\mathrm{TrueLeastOptimal}}}$$
where a PSR of 100 implies the least optimal dose was chosen, and a PSR of 0 implies that the optimal dose was chosen.

##### Inaccuracy

Inaccuracy in this setting was defined by the predicted utility score of the predicted optimal dose compared to the true utility at that dose. This is shown in Figure 6b and given by the following formula:(7)$$\mathrm{Inaccuracy}={\;\mathrm{PredictedUtility}}_{\mathrm{Chosen}}-{\;\mathrm{Utility}}_{\mathrm{Chosen}}$$

Ideally, this should be as close to zero as possible, which is equivalent to minimising the metric of absolute inaccuracy, which is given by:(8)Absolute Inaccuracy = max(Inaccuracy,−Inaccuracy)

##### Average Regret

Each trial individual experiences a certain level of utility from receiving a vaccine. This utility can be subtracted from the true optimal utility to determine the ‘regret’ for that individual. Average regret in this setting was defined by the utility that the average trial individual experienced relative to the true utility at the true optimal dose. Ideally, this should be minimised. This is shown in Figure 6c and given by the following formula:(9)Average Regret = Cumulative Regret / n
where n is the number of trial participants and
(10)$$\mathrm{Cumulative}\;\mathrm{Regret}=\sum_{\mathrm{individual}=1}^{n}{\mathrm{Regret}}_{\mathrm{Individual}}$$
where
(11)$${\mathrm{Regret}}_{\mathrm{Individual}}={\;\mathrm{Utility}}_{\mathrm{Individual}}-{\;\mathrm{Utility}}_{\mathrm{True}\;\mathrm{Optimal}}$$

We further defined percentage average regret to again enable comparison between scenarios.
(12)$$\mathrm{Percentage}\;\mathrm{Average}\;\mathrm{Regret}=\frac{\mathrm{Average}\;\mathrm{Regret}}{{\mathrm{Utility}}_{\mathrm{True}\;\mathrm{Optimal}\;}-{\mathrm{Utility}}_{\mathrm{True}\;\mathrm{Least}\;\mathrm{Optimal}}}$$

### 2.7. Objective 2: When Trial Size Is Fixed, How Dose-Optimisation Approaches Are Affected by the Assumed Statistical Efficacy Model and Method of Trial Dose Selection

For this objective, we assessed different methods of trial dose selection in combination with the three efficacy models. In addition to the full uniform exploration described in objective 1, which was retrospective, we considered three continual-modelling-based methods of trial dose selection. Using the definition of a dose-optimisation approach outlined above, we investigated the following approaches:i.Efficacy model: saturating, peaking, or weighted;ii.Trial size: 30;iii.Trial dose-selection method: full uniform exploration, standard fully continual modelling, balanced exploration (softmax) fully continual modelling, or three-stage (softmax).

Hence, there were 12 (=4 × 3) dose-optimisation approaches, reflecting a combination of the 4 methods of trial dose selection and 3 assumed efficacy models. Each scenario/approach pairing was simulated 100 times for a total of 16800 (=12 × 14 × 100) simulated trials and 840,000 simulated individuals.

Although the ‘full uniform exploration’ trial design assessed in objective 1 seemed a reasonable design for improving model calibration, there are drawbacks to this design. Many individuals may be trialled with a suboptimal dose due to the uniform nature of the design. Modelling is also performed retrospectively; therefore, the generated data are not used to improve trial dosing. Hence, for this objective, we considered approaches that use continual-modelling-based methods of trial dose selection, which have been proposed to lead to more ethical trials [13]. These essentially repeat a cycle of:Conducting a small trial on a select set of doses;Gathering efficacy and toxicity data from this experiment;Updating the efficacy and toxicity models based on these data;Using the models to select either the next set of doses to test or to select the final dose to predict as ‘optimal’.

#### 2.7.1. Fully Continual Standard

The standard fully continual method is the simplest continual modelling dose-selection method. Each ‘experiment’ consists of one individual tested with the model-predicted optimal dose. 

#### 2.7.2. Fully Continual, Balanced Exploration (Softmax)

The standard fully continual dose-selection method above has previously been shown to be potentially useful in drug dose optimisation; however, analysis of optimisation problems outside of dose finding have shown that testing only the predicted optimal may not be beneficial [42]. Being willing to ‘explore’ doses that are not predicted to be optimal may ultimately improve the final selected dose. As such, we considered softmax selection [43,44], where doses with higher predicted utilities were more likely to be selected; however, the selected trial doses were not always exactly at the predicted optimal. The degree of exploration was controlled by an exploration parameter, and further detail is given in Appendix A.

#### 2.7.3. Three-Stage (Softmax)

Whereas the fully continual modelling process has been shown to be effective in drug dose optimisation, typically in that setting, the time between treatment and measurement of effect is short. In the vaccine setting, the time between vaccination and measurement of effect (immunological response) could be days, weeks, months, or even years. Hence, the application of a fully continual modelling process could take much longer than is feasible. We therefore considered a dose-selection method that contained elements of both the fully continual and fully retrospective modelling designs.

There are many ways this could be implemented. We considered a three-stage approach as follows:Stage 1.
a.⅓ of the trial population is dosed following the full uniform exploration approach outlined in objective 1.b.Efficacy and toxicity models are calibrated using these data and pseudo-data [3.7.5].Stage 2.
a.The second ⅓ of the population is dosed according to the utility predictions of the combined efficacy and toxicity models, using the softmax selection method with relatively high exploration.b.Efficacy and toxicity models are calibrated using these data, data from step one, and downweighted pseudo-data.Stage 3.
a.The final ⅓ of the population is dosed according to the utility predictions of the combined efficacy and toxicity models, using the softmax selection method with relatively low exploration.b.Efficacy and toxicity models are calibrated using all collected data, with pseudo-data being ignored. The predicted optimal dose is selected according to the utility predictions of the combined efficacy and toxicity models.

#### 2.7.4. Dose-Escalation/De-Escalation Rules

We also included a simple escalation/de-escalation rule for the fully continual dose-selection methods, which is typically suggested for such continual modelling dose-selection methods. The first dose was always 5 on the log_10_ scale (that is to say the middle dose). A dose could not be in excess of ½ a log above of the maximum previously tested dose or more than ½ a log below the minimum previously tested dose. For example, dose 10 (10^10^) could not be tested unless a dose of at least 9.5 (10^9.5^) had been previously tested. This was suggested to reduce the risk of unexpected higher-grade toxicities.

As the first stage of the three-stage softmax approach included the smallest and largest allowed doses in the dosing space, the dose escalation/de-escalation rules would have no effect.

#### 2.7.5. Pseudo-Data

Such continual modelling approaches can be implemented when insufficient data are available. Calibration with a small amount of data can be unstable; hence, pseudo-data were used to stabilise the calibration, as suggested in [15]. We used minimally informative pseudo-data, which was quickly outweighed by real data and was ignored in the calibration step prior to final dose selection. Full details can be found in Appendix A.

#### 2.7.6. Comparison between Approaches/Trial Designs

As in objective 1, percentage simple regret, inaccuracy, absolute inaccuracy, average regret, and percentage average regret were calculated. We used the Copeland method to identify a quantitative ranking of these approaches for their simple regret, absolute inaccuracy, and average regret outcomes [45,46], see Appendix A. Sum of ranks and mean of Copeland metrics across simple regret, absolute inaccuracy, and average regret were also obtained.

## 3. Results

### 3.1. Objective 1: When the Method of Trial Dose Selection Is Fixed, How Dose-Optimisation Approaches Are Affected by the Assumed Statistical Efficacy Model and Trial Size

A clear relationship between trial size and percentage simple regret (PSR) was observed (Figure 7), with a reduction in PSR as trial size increased, indicating that a more optimal dose was selected when trial size was larger. This was true regardless of whether a saturating, peaking, or weighted efficacy model was used and suggests an increased trial size improved final dose selection. However, the PSR aggregated across all scenarios was lower for the peaking and weighted approaches than for the approaches with saturating efficacy models (Figure 7), suggesting that using either a peaking or weighted model increased the average utility of the final selected dose. For almost all scenarios and trial sizes, it was better to assume a peaking curve than a saturating curve to minimise PSR, with a few exceptions, see Appendix A.

Similarly, the accuracy of the predicted utility at the predicted optimal dose increased with increasing trial size (decreased inaccuracy) (Figure 8). An ‘optimistic bias’ was observed (positive inaccuracy) (Figure 8a), with predicted utility typically higher than the true utility for the given dose, as is often expected in optimisation problems [48,49] (Appendix A). There was no difference in inaccuracy between efficacy models for any trial size.

There was no difference in median percentage average regret between efficacy model and trial size (Figure 9). This was to be expected, as all used the same method of trial dose selection of full uniform exploration with no continual modelling to allow later trial participants to benefit from early trial data.

All plots for PSR, inaccuracy, and percentage average regret for each scenario are shown in Appendix A. Analysis of the distributions of PSR, absolute inaccuracy, and PAR for statistical significance are given in the Appendix A.

### 3.2. Objective 2: When Trial Size Is Fixed, How Dose-Optimisation Approaches Are Affected by the Assumed Statistical Efficacy Model and Method of Trial Dose Selection

#### 3.2.1. Qualitative Analysis

With a trial of size 30, we found that using the peaking or weighted efficacy model still typically led to more optimal dose selection when compared to the saturating model (as shown by decreased PSR) (Figure 10). Neither the full uniform exploration modelling approaches nor the continual modelling approaches consistently showed a reduced PSR relative to one another. For some scenarios (saturating 5, Appendix A) we found that PSR was reduced by using continual modelling approaches. For others (peaking 1, peaking efficacy curve assumed, Appendix A), we found that the full uniform exploration approach appeared to best reduce PSR. This may suggest that the benefits of high levels of exploration or continual modelling for reducing PSR depend on the scenario. In general, the fully continual balanced exploration modelling approaches and the three-stage softmax approach appeared to lead to a slight reduction in PSR across the 14 scenarios relative to the standard fully continual modelling approach, suggesting that exploration may be important in consistent dose optimisation.

With a trial size of 30, there was minimal difference in inaccuracy and absolute inaccuracy across the approaches (Figure 11). This may suggest that the accuracy of utility predictions at the model predicted optimal vaccine dose was not dramatically improved by using a continual modelling method of dose selection. There was still an optimistic bias, although this was slightly reduced in the three-stage approaches relative to the standard fully continual, balanced exploration fully continual, and full uniform exploration approaches. Again, this was minimal relative to the differences that were observed when changing trial size in objective 1.

With a trial size of 30, the results suggest that fully continual modelling (both standard and balanced) and three-stage approaches identify optimal dose with a greater net benefit to trial participants than the retrospective full uniform exploration approaches (as shown by decreased average regret) (Figure 12). The balanced exploration variant of the fully continual modelling dose-selection method appeared to have a marginally increased percentage average regret compared to approaches with standard fully continual modelling dose selection, but average regret was still significantly reduced relative to approaches using the three-stage softmax or full uniform exploration methods of trial dose selection. The three-stage softmax approaches showed a reduced average regret relative to full uniform exploration but a greater average regret relative to the fully continual approaches. These findings were the same regardless of the assumed efficacy model.

Similar plots for each individual scenario are shown in Appendix A. Analysis of the distributions of PSR, absolute inaccuracy, and PAR for statistical significance are given in the Appendix A.

#### 3.2.2. Quantitative Ranking

For minimising PSR, the approach assuming a weighted efficacy curve and using fully continual modelling with balanced exploration was most consistent across the scenarios that we tested (Table 3). The fully continual modelling with balanced exploration approaches outranked the standard fully continual modelling approaches for each efficacy model. The three-stage softmax approaches also performed well, along with the approach with full uniform exploration with an assumed peaking efficacy curve. This may suggest that when assuming a peaking curve shape, exploration improves final dose selection.

For minimising average regret, the standard fully continual modelling approach assuming a peaking efficacy curve was most consistent across the scenarios that we tested. The shape of the model’s efficacy curve was less important than the method of trial dose selection for minimising average regret, with the order from worst to best being full uniform exploration, three stage softmax, balanced fully continual modelling, and standard fully continual modelling. This may suggest that for small trial sizes (30), the standard fully continual modelling approach is most ethical, as the average regret was lowest. Therefore, the reduction in simple regret observed when including exploration may come at the cost of increased average regret for such small trial sizes.

For minimising absolute inaccuracy, the three-stage softmax approach assuming a peaking efficacy curve was most consistent across these scenarios, suggesting that dosing trial participants both near the predicted optimal and further away from the predicted optimal may reduce inaccuracy. The full uniform exploration approaches ranked lowest.

The dose-optimisation approach with an assumed weighted efficacy curve and fully continual modelling with balanced exploration had the best sum of ranks, which suggests that this approach should be chosen if simple regret, inaccuracy, and average regret are all equally valued. Copeland tables for each scenario are given in Appendix A.

## 4. Discussion

In this work, we used simulation studies to evaluate mathematical-modelling-based approaches to optimising vaccine dose, maximising efficacy while minimising toxicity. We found that doses selected using these methods were improved with increased trial size, particularly for studies with at least 30 trial participants. Using a peaking model or a weighted model averaging approach for modelling dose efficacy was generally most effective for determining optimal dose. Identification of optimal dose was minimally affected by the method of trial dose selection. However, using modelling at interim timepoints to select trial doses led to trial participants receiving more optimal doses. Dosing only at the predicted optimal dose during a clinical trial may lead to less optimal dose selection relative to dosing around the predicted optimal. This work suggests modelling can be used effectively for vaccine dose finding, accelerating effective vaccine development and saving lives.

There were a number of strengths in our work. We included ordinal toxicity, which is highly relevant in vaccines due to their general safety profiles and potential for prophylactic use. Whereas we have previously seen vaccine dose optimisation applied using real-world data [3,24], simulation studies allow for an increased understanding of the potentials and pitfalls of dose-finding methodologies because the “truth” is known. By explicitly defining scenarios, we were able to accurately test metrics such as PSR and inaccuracy for these dose-optimisation approaches. Additionally, the scenarios we chose explored a wide range of curve shapes so that a multitude of potential real-life dosing scenarios could be reflected.

We chose to consider optimisation over a large number of potential doses, whereas previous dose-optimisation simulation studies have typically focussed on choosing between a small number of dosing levels [50]. Using a small number of doses may not be appropriate if none of the selected doses achieves optimal vaccine utility. Additionally, we chose to use ‘simple regret’-based metrics rather than ‘percentage best arm identification’, which considers an approach to have been successful in optimising dose in a simulated clinical trial if and only if the true optimal dose was predicted to be optimal [51]. Using ‘simple regret’ as a metric is more appropriate for vaccine dose-optimisation, as multiple closely spaced arms may have similar utility. Selecting a dose with approximately equal clinical value to that of the true optimal dose would be considered to be an improvement to selecting an inferior dose under simple regret, but both would be considered as failures in terms of optimising dose under the percentage best arm identification metric.

Additionally, we discussed the concept of ‘exploration’, which is rarely considered in dose-optimisation work but is instrumental to the wider class of ‘multi-armed bandit problems’, which dose-optimisation can be considered to be part of [52]. The analysis of full uniform exploration and weighted modelling approaches was also novel in this setting. Finally, we also evaluated the concept of accuracy and inaccuracy in dose-optimisation modelling approaches, which is typically not well researched. This seems relevant, given the overestimation bias observed across all of these approaches to dose optimisation, which could lead to overestimations in the potential of vaccine utility and incorrectly guide policy.

There were weaknesses with both this work and the dose-optimisation approaches evaluated in general. It is likely that the 14 scenarios we chose may not represent all possible real-life dose–efficacy, dose–toxicity, and dose–utility curves. However, the 14 scenarios we used were qualitatively different and sufficient to cover plausible prior belief for any specific vaccine. When designing dose-finding trials for a future vaccine, it may be reasonable to consider only the findings for scenarios that are most similar to the clinician’s prior beliefs about a given vaccine’s likely dose–efficacy/dose–toxicity curves. Additionally, we performed only 100 clinical trial simulations for each approach/scenario pairing. This is in excess of the minimum of 10 that has been suggested [53], and we believe a larger number of simulated trials would not have impacted the results. Although not a weakness of the work, the observed overestimation bias appears to be a weakness of these optimisation approaches, as it decreases with increased trial size. We note that an overestimation bias is expected in both model-based [48,49] and traditional comparative dose-selection [54] methodologies (Appendix A). Methods to remedy this have been suggested but were not addressed in this work [49,55].

Our work is consistent with previous modelling findings. These studies also showed that continual modelling approaches increased the average quality of clinical trial drug dosing (e.g., decreased average regret) [12]. It has previously been shown that for similar optimisation problems, exploration is key to maximising utility, but the level of exploration varies based on scenario and sample size [56]. This is consistent with our finding that for a sample size of 30, whether approaches without exploration could outperform explorative approaches depended on the scenario. Specifically, in previous work on optimisation of drug dose (using a small number of potential dosing levels, Bayesian methodologies, and only an assumed saturating efficacy curve), it was found that including exploration was beneficial for drug dose optimisation [57]. We also found, with regards to efficacy curves, that the peaking latent quadratic curve often outperformed the saturating sigmoid curve in cases where the true scenario curve was sigmoid saturating. Although we might expect that using the model that best describes true dose–response dynamics would be preferable for optimisation purposes, previous studies suggested that in some cases, models that do not well approximate the true dynamics can be preferable for optimisation purposes [58].

There were limitations to this work and the approaches discussed. We assumed that a binary measure for vaccine efficacy is known; however, for many diseases, a surrogate (binary or otherwise) of protection is not known. We also excluded many complicating factors that have been discussed in previous continual modelling literature. For example correlation in the probabilities of efficacy and toxicity [59], multiple toxicity subtypes (e.g., pain and nausea) [60], stopping rules [13], and placebo doses [61] were not considered. Given that modelling-based vaccine dose optimisation is a large topic that is still in the proof-of-concept stage, these were omitted to simplify the work. Additionally, we did not address cost and time requirements for trials. The time taken by continual modelling approaches in a vaccine clinical trial setting may not be justified by the resulting improvement in outcome for trial participants, which reflected by a decrease in average regret. We also did not compare these approaches to model-free approaches, such as 3 + 3 design [62,63], as such approaches are inherently designed to choose between a small number of doses and therefore have similar problems associated with selecting from a small number of doses that we discussed above.

We also did not include a fifth or sixth toxicity grade, which would typically represent a serious adverse event resulting in hospitalisation or death, respectively. As these events are likely to be rare in most vaccine trials [1] and would typically require the trial to be stopped, we excluded these gradings. Finally, we did not consider potential confounders (for example, age or sex) that may occur in practical dose-ranging trials. This would have further increased the complexity of this work.

There is much future work to be done on this topic. Additional scenarios should be tested to investigate any further shortcomings of these approaches. Only one simple utility function was considered in this work. This could be made more complex by including dose–cost relationships or modelling dose–response curves for multiple different efficacy or toxicity responses [11,60,64]. Creating a meaningful utility function is non-trivial [65,66] and is key to effective optimisation. Previous work has shown that both Bayesian methodologies and the frequentist methodologies discussed in this work perform similarly for some continual modelling approaches [67], but this should be further tested and validated. Optimising the degree of exploration that should occur could potentially decrease simple regret and average regret, but the optimal amount of exploration almost certainly depends on the scenario and trial size [56]. Additionally, using mechanistic models for dose efficacy could be beneficial if there is a good understanding of the immunodynamics relating to the vaccine [3,4,68]. However, this would likely introduce more complexity to the modelling process and to the utility function.

Although simulation demonstrates that these dose-optimisation approaches could be used when designing trials to optimise vaccine dose, these approaches clearly need to be tested in a real-world setting to evaluate their practical implementation. Although the continual modelling approaches reduced average regret (improved trial participant outcomes) relative to the retrospective approaches, there is clearly a trade-off regarding whether this is worth the increased time requirements (which dramatically increase when using fully continual approaches) or additional complexity (particularly in approaches that use softmax selection). Hence, when using modelling for vaccine dose optimisation, there appears to be a balance between improved trial participant experience and the cost of increased time to clinical use. Whether this is ethically justified is a matter for further discussion. There may also be discussion of whether the potential for greater information efficiency from modelling may reduce trial size relative to standard dose-finding trial design and justify the increased time requirements. Furthermore, there should be consideration of how to approach confounding variables. Clinical trial design randomisation typically aims to ensure that populations in different dosing groups are homogenous [69]. The approaches discussed here assume that individuals are independent of each other; therefore, randomised sampling from a homogenous population should still be used to minimise risk of confounding variables (for example, avoiding correlation between dose and age of trial participants). Additionally, choosing optimal dose for prime-boost paradigm vaccines may require more complicated mathematical modelling methods, as efficacy and toxicity outcomes may be dependent on both prime and boost dose [70].

In drug development, mathematical modelling methodologies have led to improved drug efficacy and toxicity profiles, as well as a reduction in the cost of clinical trials. Despite the limitations and open questions discussed above, the application of mathematical modelling methodologies in vaccine pharmacological and biotechnology industries could allow for more quantitative and informed decision making.

## 5. Conclusions

Choosing the optimal vaccine dose is a complicated endeavour. Through this work, we evaluated model-based dose-optimisation approaches, along with trial design, to utilise these methodologies. Model-based dose-optimisation approaches may be effective for making vaccine dose decisions, which may increase efficacy and decrease toxicity, both during clinical trials and upon vaccine implementation. We hope that this work leads to future research and practical application of modelling methods in selecting vaccine doses. This may accelerate effective vaccine development and save lives.

## Figures and Tables

**Figure 1 vaccines-10-00756-f001:**
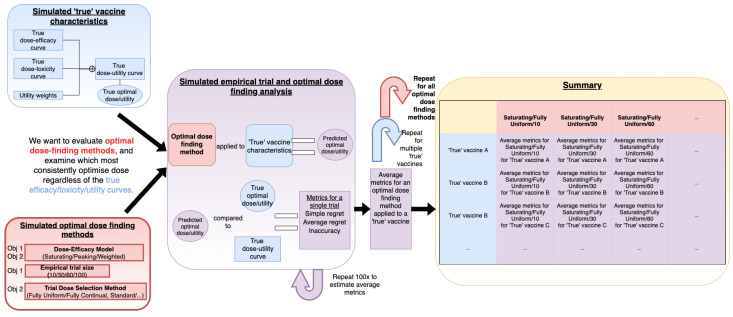
Visual depiction of the process of conducting simulation studies used in this work to assess mathematical-modelling-based dose-optimisation approaches. The aim was to evaluate dose-optimisation approaches (red), in particular the effect of changing the assumed dose–efficacy model, trial size, and trial dose-selection method. These were tested by simulating clinical trials (purple) based on ‘scenarios’ (blue). Repeated simulation of clinical trials was conducted for different dose-optimisation approach/scenario pairs, and metrics related to how effectively optimal dose was located were calculated. These were tabulated and compared to assess whether the assumed dose–efficacy model, trial size, and trial dose-selection method influence the consistency of dose optimsation.

**Figure 2 vaccines-10-00756-f002:**
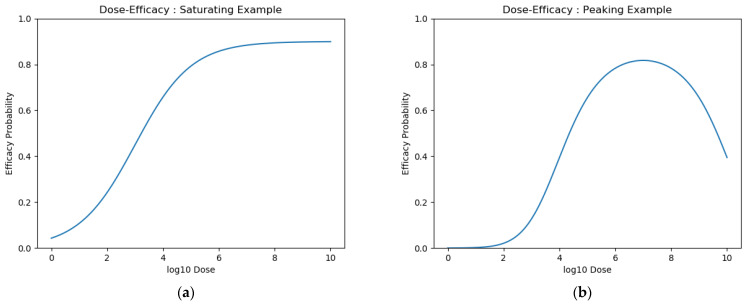
Example curves for (**a**) saturating and (**b**) peaking dose efficacy.

**Figure 3 vaccines-10-00756-f003:**
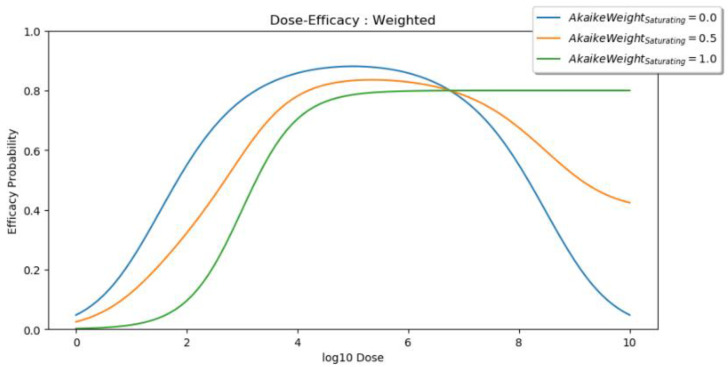
Visual example of model averaging. When the saturating Akaike weight is 0, the predicted efficacy curve is defined entirely by the peaking model (blue). When the saturating Akaike weight is 1, the predicted efficacy curve is defined entirely by the saturating model (green). If both models are equally as likely, given the available data, then the saturating Akaike weight and the peaking Akaike weight are both 0.5, and the predicted efficacy curve is the midpoint of the saturating and peaking curves (orange).

**Figure 4 vaccines-10-00756-f004:**
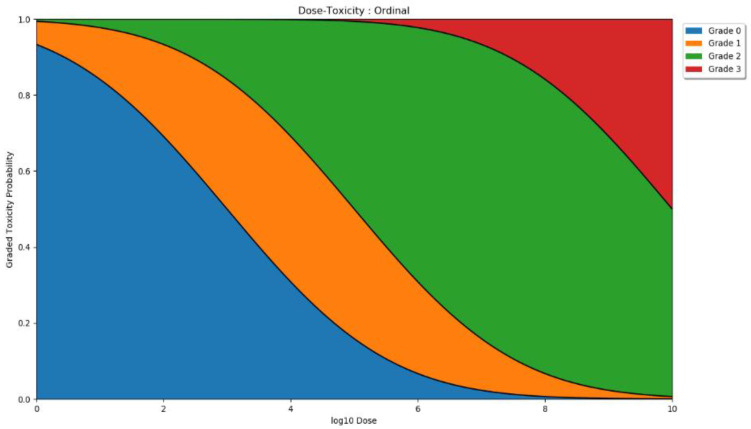
Visual example of ordinal dose toxicity. The plot shows the proportion of individuals that would experience different adverse event grades for each dose. In this example, at low doses, grade 0 (blue) adverse events are most likely. By dose 6, grade 1 (yellow) and grade 2 (green) adverse events are likely but grades 0 and 3 are also possible. By the maximum dose, approximately 50% of individuals would experience a grade 3 adverse event, and almost all others would experience grade 2 events.

**Figure 5 vaccines-10-00756-f005:**
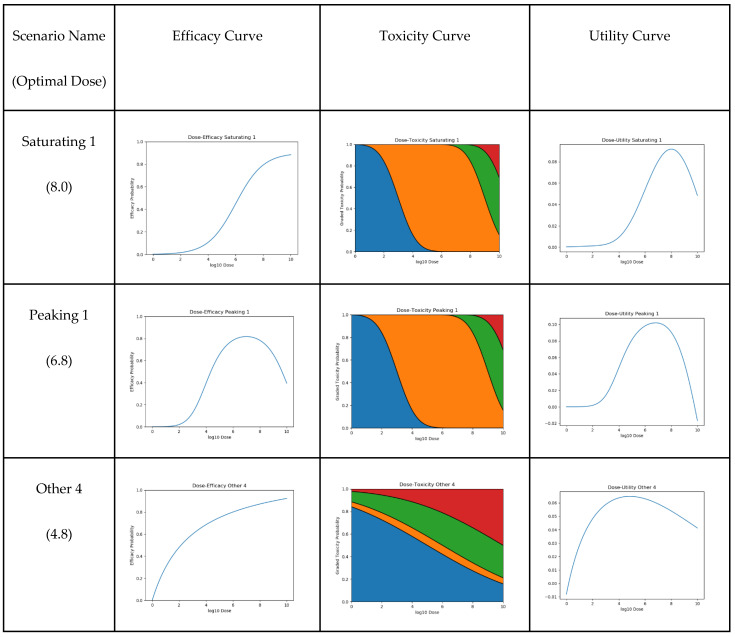
Three examples of the 14 tested scenarios. For each scenario, we show dose efficacy, dose toxicity, and the resultant dose–utility plots. Optimal dose is also given. For the toxicity plots, grade 0, 1, 2, and 3 adverse event probabilities are represented by blue, orange, green, and red, respectively.

**Figure 6 vaccines-10-00756-f006:**
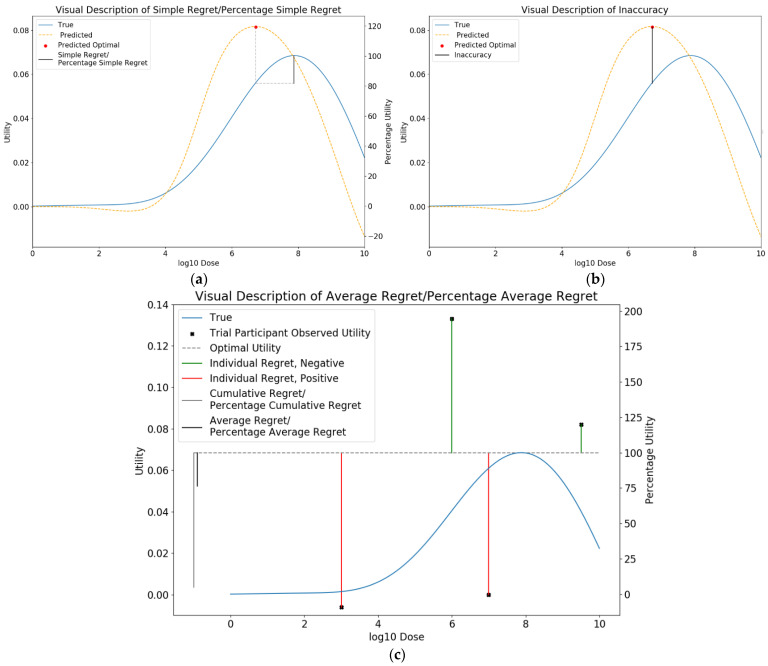
Visual description of (**a**) simple regret, (**b**), inaccuracy, and (**c**) average regret.

**Figure 7 vaccines-10-00756-f007:**
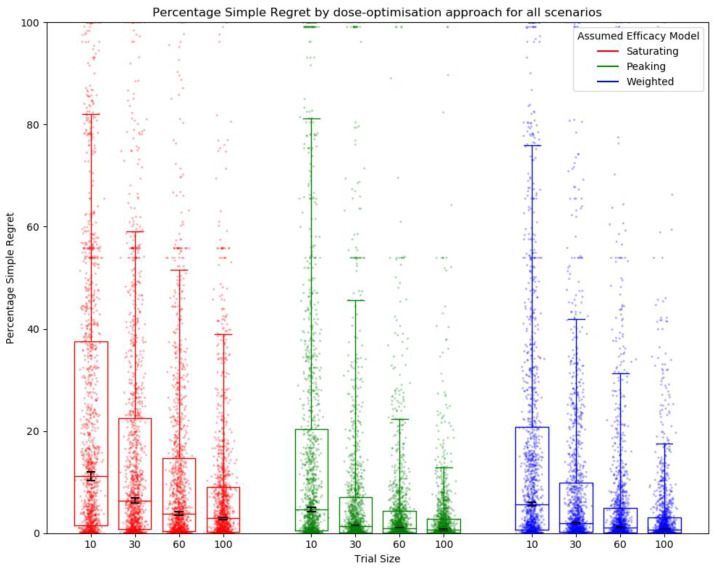
Percentage simple regret (PSR) for all scenarios by assumed efficacy model and trial size. Trial dose selection method was full uniform exploration. A lower PSR denotes a more optimal final dose. Individual points represent PSR for a single simulated clinical trial using one dose-optimisation approach for one of the 14 scenarios. The middle line of each boxplot is the median value; the box marks the 25th and 75th percentiles, and the whiskers mark the 5th and 95th percentiles of the data. Black lines represent the 95% confidence interval for the median of each distribution [47]. The majority of these distributions of PSR were different to a statistically significant extent at the *p* = 0.05 threshold according to the Kolmogorov–Smirnov test due to the large number of simulations conducted (100 per approach/scenario pairing). For further details on statistical significance see Appendix A.

**Figure 8 vaccines-10-00756-f008:**
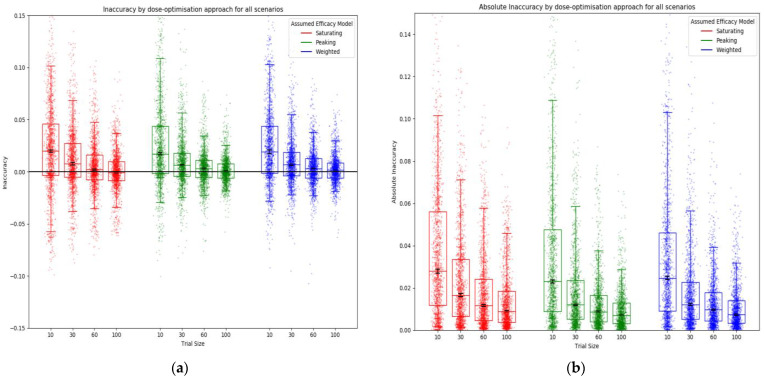
Inaccuracy (**a**) and absolute inaccuracy (**b**) for all scenarios by assumed efficacy model and trial size. Trial dose-selection method was full uniform exploration. The closer the inaccuracy/absolute accuracy was to 0, the more accurate the prediction of utility was at the predicted optimal dose. Individual points represent inaccuracy/absolute inaccuracy for a single simulated clinical trial using that dose-optimisation approach for one of the 14 scenarios. The middle line of each boxplot is the median value; the box marks the 25th and 75th percentiles, and the whiskers mark the 5th and 95th percentiles of the data. Black lines represent the 95% confidence interval for the median of each distribution [47]. The majority of these distributions of absolute inaccuracy were different to a statistically significant extent at the *p* = 0.05 threshold according to the Kolmogorov–Smirnov test due to the large number of simulations conducted (100 per approach/scenario pairing). For further details on statistical significance, see Appendix A.

**Figure 9 vaccines-10-00756-f009:**
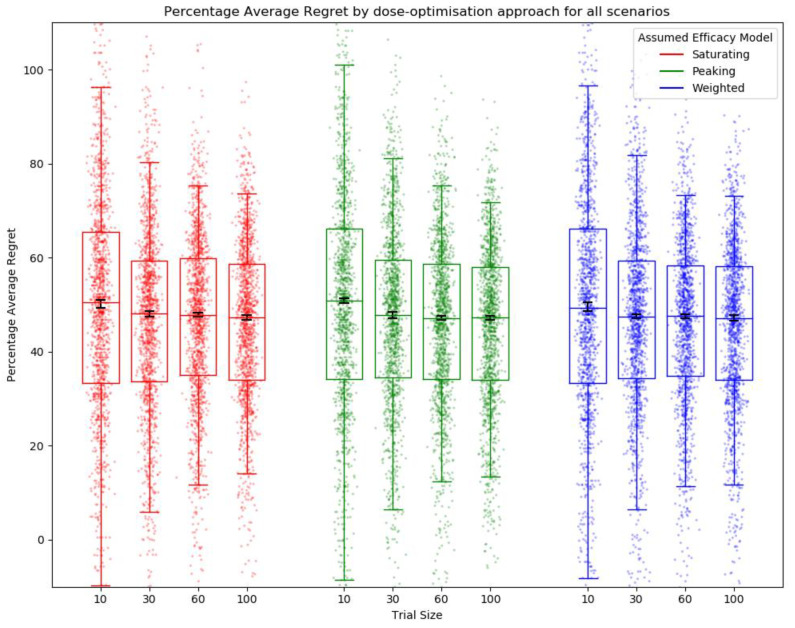
Percentage average regret for all scenarios by assumed efficacy model and trial size. Trial dose-selection method was full uniform exploration. Individual points represent percentage average regret for a single simulated clinical trial using that dose-optimisation approach for one of the 14 scenarios. The middle line of each boxplot is the median value; the box marks the 25th and 75th percentiles, and the whiskers mark the 5th and 95th percentiles of the data. Black lines represent the 95% confidence interval for the median of each distribution [47]. The majority of these distributions of PAR were not different to a statistically significant extent at the *p* = 0.05 threshold according to the Kolmogorov–Smirnov test. For further details on statistical significance, see Appendix A.

**Figure 10 vaccines-10-00756-f010:**
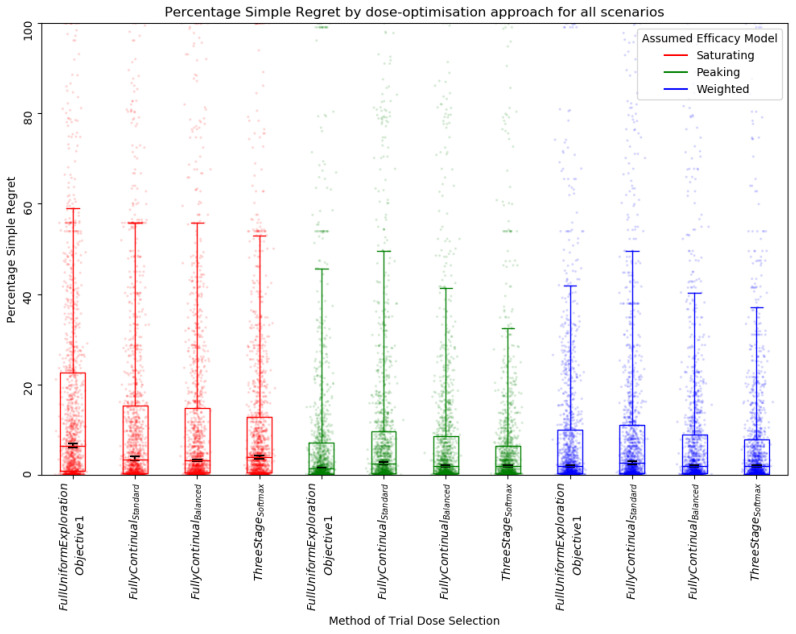
Percentage simple regret (PSR) for all scenarios by assumed efficacy model and trial dose-selection method. Trial size was 30. Individual points represent PSR for a single simulated clinical trial using that dose-optimisation approach for one of the 14 scenarios. The middle line of each boxplot is the median value; the box marks the 25th and 75th percentiles, and the whiskers mark the 5th and 95th percentiles of the data. A lower PSR denotes a more optimal final dose. Black lines represent the 95% confidence interval for the median of each distribution [47]. The distributions of PSR for the approaches that assumed a saturating model were different to the distributions of the approaches that assumed a peaking or weighted efficacy mode to a statistically significant extent at the *p* = 0.05 threshold according to the Kolmogorov–Smirnov test. For further details on statistical significance, see Appendix A.

**Figure 11 vaccines-10-00756-f011:**
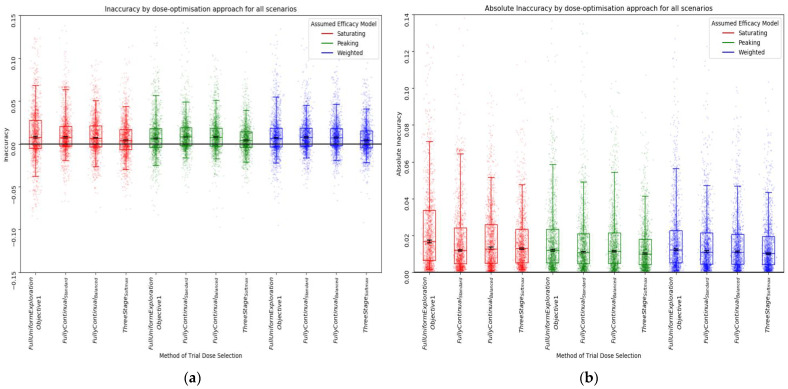
Inaccuracy (**a**) and absolute inaccuracy (**b**) for all scenarios by assumed efficacy model and trial dose-selection method. Trial size was 30. Individual points represent inaccuracy/absolute inaccuracy for a single simulated clinical trial using that dose-optimisation approach for one of the 14 scenarios. The middle line of each boxplot is the median value; the box marks the 25th and 75th percentiles, and the whiskers mark the 5th and 95th percentiles of the data. The closer inaccuracy/absolute accuracy is to 0, the more accurate the prediction of utility is at the predicted optimal dose. Black lines represent the 95% confidence interval for the median of each distribution [47]. The majority of these distributions of absolute inaccuracy were not different to a statistically significant extent at the *p* = 0.05 threshold according to the Kolmogorov–Smirnov test. For further details on statistical significance, see Appendix A.

**Figure 12 vaccines-10-00756-f012:**
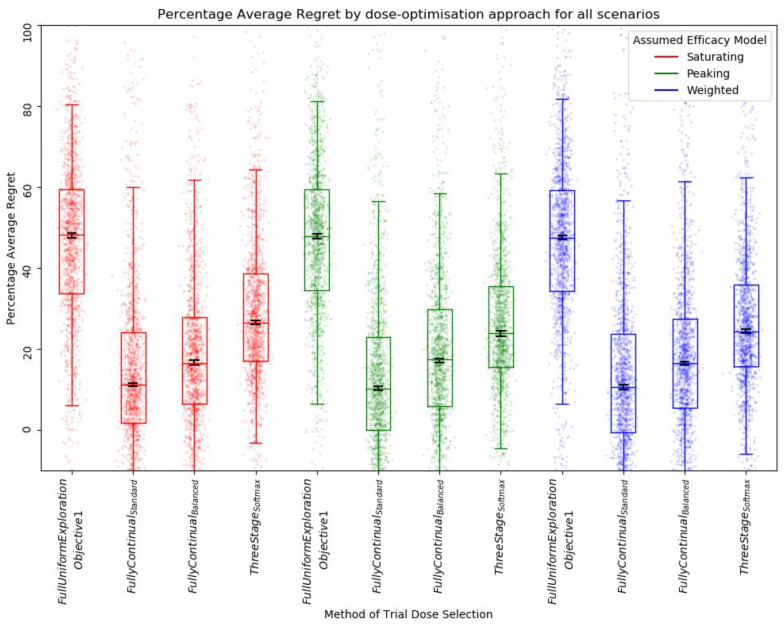
Percentage average regret for all scenarios by assumed efficacy model and trial dose-selection method. Trial size was 30. Individual points represent percentage average regret for a single simulated clinical trial using that dose-optimisation approach for one of the 14 scenarios. The middle line of each boxplot is the median value; the box marks the 25th and 75th percentiles, and the whiskers mark the 5th and 95th percentiles of the data. A lower percentage average regret denotes better outcomes for trial participants. Black lines represent the 95% confidence interval for the median of each distribution [47]. The majority of these distributions of PAR were not different to a statistically significant extent at the *p* = 0.05 threshold according to the Kolmogorov–Smirnov test. For further details on statistical significance, see Appendix A.

**Table 1 vaccines-10-00756-t001:** Description of the assumed grades of adverse events. These follow the gradings described in [30,31].

Adverse Reaction Grade	General Description
0	None.
1	Mild. Does not interfere with normal activity.
2	Moderate. Interference with normal activity. Little or no treatment required.
3	Severe. Prevents normal activity. Requires treatment.

**Table 3 vaccines-10-00756-t003:** Copeland scores and rankings for all approaches with a trial size of 30 across all scenarios. Ordering is by aggregate rank. Aggregate rank was calculated as the sum of ranks for simple regret, absolute inaccuracy, and average regret. Aggregate score was the mean of scores for simple regret, inaccuracy, and average regret.

	Aggregate of Simple Regret, Absolute Inaccuracy, and Average Regret		Simple Regret		Absolute Inaccuracy		Average Regret	
Approach	Rank	Score	Rank	Score	Rank	Score	Rank	Score
Weighted, Fully Continual, Balanced	8	0.570	1	0.564	3	0.522	4	0.625
Peaking, Fully Continual, Standard	12	0.572	7	0.498	4	0.517	1	0.701
Peaking, Softmax Three Stage	12	0.536	4	0.552 ^1^	1	0.556	7	0.500
Peaking, Fully Continual, Balanced	14	0.557	3	0.552 ^1^	6	0.510	5	0.610
Weighted, Fully Continual, Standard	15	0.565	8	0.485	5	0.514	2	0.698
Weighted, Softmax Three Stage	15	0.528	5	0.541	2	0.549	8	0.493
Saturating, Fully Continual, Standard	20	0.543	10	0.447	7	0.492	3	0.691
Peaking, Full uniform exploration	24	0.414	2	0.563	10	0.480	12	0.201
Saturating, Fully Continual, Balanced	24	0.519	9	0.463	9	0.486	6	0.609
Saturating, Softmax Three Stage	28	0.465	11	0.442	8	0.489	9	0.465
Weighted, Full uniform exploration	28	0.400	6	0.516	11	0.480	11	0.203
Saturating, Full uniform exploration	34	0.330	12	0.378	12	0.406	10	0.205

^1^ Scores are rounded to three decimal places, but ranks were calculated before rounding.

## Data Availability

Data and code for this work are available through https://github.com/ISIDLSHTM/Model_Size_Selection (accessed on 10 April 2022).

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
