# Peer review of "Mathematical Modelling for Optimal Vaccine Dose Finding: Maximising Efficacy and Minimising Toxicity"

_vaccines, 2022, doi:10.3390/vaccines10050756_

Round 1

Reviewer 1 Report

The topic submitted is novel and adds significant research data to the existing field of research in vaccines and related therapy.

The article is not very well articulated and needs English language revisions and even formatting of the manuscript as per the MDPI guidelines.

The manuscript needs to be checked for statistical significance, especially in fig 8,9,10. The introduction needs to be concise.

Abstract and conclusion should include a sentence proposing the future direction of the present mathematical model.

Also the commercial aspects of how pharma or biotech industries can benefit from using the m.model. 

Most of the methodology lacks citing the references from where the technique was adopted and used to conduct the study.

Below are a few suggestions the authors are requested to discuss and cite in appropriate sections.

Journal of theoretical biology465, pp.51-55.

 Molecules 27, no. 4 (2022): 1170.

Npj Vaccines 3, no. 1 (2018): 1-7.

The Lancet Infectious Diseases 21, no. 6 (2021): 793-802.

Author Response

Please see the attachment to see the figures referenced.

The topic submitted is novel and adds significant research data to the existing field of research in vaccines and related therapy.

We thank the reviewer for their time and expertise in reviewing this work.

The article is not very well articulated and needs English language revisions and even formatting of the manuscript as per the MDPI guidelines.

Thank you. The article has been reformatted to be in line with MDPI guidelines in the version of the manuscript that has been requested to be amended on the MDPI portal. 

We have read through the article and made English language revisions. 

The manuscript needs to be checked for statistical significance, especially in fig 8,9,10.

Thank you. We agree that discussion of statistical significance, usually by a p-value threshold, is common in vaccine literature and may be preferable to some readers. However this is less common in simulation studies for dose-response modelling [for examples see [16,17]], as typically the large number of simulations in such studies means that even small differences in distributions become ‘significantly different’. We considered a potential visualisation of  statistical significance for figure 8b, Objective 2 Absolute Inaccuracy, with a red bar indicating that two distributions were significantly different by the Kolmogorov–Smirnov Test at the p=0.05 threshold.   

Please see the attachment for the figure.

We hope that the reviewer agrees that this does not improve comprehension of the results, and that the additional complexity in the figure is distracting and potentially confusing. However, we do acknowledge and agree that it is important to show that our findings are not down to chance and that our choice of 100 simulations was sufficiently large.  We hence added black lines to show the 95% confidence interval for each of the median values, which we believe justifies that these median values are precise enough to draw meaningful inference and also visually showing where significance should be expected (which was for nearly all results). Below we show an example of this, for figure 8 again.

Please see the attachment for the figure.

Further to this, we added a discussion of statistical significance into the supplementary, which we agree may be of interest to some readers. We have also added text to the captions of these figures discussing statistical significance and directing readers to the supplementary for further detail in case they desire this information. 

16. James, G.D.; Symeonides, S.; Marshall, J.; Young, J.; Clack, G. Assessment of Various Continual Reassessment Method Models for Dose-Escalation Phase 1 Oncology Clinical Trials : Using Real Clinical Data and Simulation Studies. BMC Cancer 2021, 21.

17. Takahashi, A.; Suzuki, T. Bayesian Optimization Design for Dose-Finding Based on Toxicity and Efficacy Outcomes in Phase I/II Clinical Trials. Pharm. Stat. 2021, 20, 422–439, doi:10.1002/pst.2085.

The introduction needs to be concise.

We thank the reviewer for this comment. We agree that being concise is beneficial, but could not find any text in the introduction that we did not believe was needed in order to provide a comprehensive introduction to these complex methods.

Abstract and conclusion should include a sentence proposing the future direction of the present mathematical model.

We have added text to the conclusion and abstract to this effect, proposing that the future direction of these mathematical models should be in practical application. We thank the reviewer for this useful suggestion.

Also the commercial aspects of how pharma or biotech industries can benefit from using the m.model.

We have added text to the discussion to suggest that the use of these models could reduce trial expense, as they have in drug development. We believe that the reviewer’s comment has improved this work, as we would support practical applications of these modelling methodologies in industry.

Most of the methodology lacks citing the references from where the technique was adopted and used to conduct the study.

We have added text to reference similar simulation studies. We thank the reviewer for this suggestion.

Below are a few suggestions the authors are requested to discuss and cite in appropriate sections.

We thank the reviewer for these suggested papers and discuss these below. 

Journal of theoretical biology, 465, pp.51-55.

We agree that this paper is relevant. We have cited this work already [reference 4], however our bibliography software defaulted to the pre-publication arxiv reference. We have corrected this to the correct reference, and thank the reviewer for noticing the seeming absence of reference to this highly-relevant paper.

 Molecules 27, no. 4 (2022): 1170.

This work on cationic lipid nanoparticles was an interesting read. We agree that the data in sections 2.5 and 2.6 represent interesting concentration-response data that could potentially be used for mathematical modelling. The authors do also discuss their choice of an ‘optimal’ concentration based on observed data. We have not chosen to include reference to this work as we have no specific expertise in nanoparticle science and the work does not (to our understanding) use mathematical dose-response modelling.

If the reviewer has further insight into the importance of the ‘Molecules 27 ‘ paper with regards to our work or with regards to the future or background of vaccine dose optimisation then we would consider further the possibility of including reference to this article.

Npj Vaccines 3, no. 1 (2018): 1-7.

The work suggested here is a useful discussion of the relevant data of efficacy and toxicity outcomes for a wide collection of COVID-19 vaccines. The dose-response data is particularly interesting. Whilst our understanding of the work is that no modelling was conducted, we believe that this work highlights some relevant discussion, particularly regarding interactions between prime and boost vaccine dose. We have therefore added the following text to the discussion.

“Additionally, choosing optimal dose for prime-boost paradigm vaccines may require more complicated mathematical modelling methods, as efficacy and toxicity outcomes may be dependent on both prime and boost dose[Npj Vaccines 3, no. 1 (2018): 1-7.] “.

The Lancet Infectious Diseases 21, no. 6 (2021): 793-802.

This work represents the use of mathematical models in describing epidemiological disease dynamics. This represents another area where mathematical modelling of biological mechanisms has been important decision making for health policy. It also highlights that having a quantitative understanding of vaccine efficacy is important for predicting population scale outcome on vaccine rollout. We have included reference to this article in the method section with the following text:

“Practically WeightEfficacy could be chosen based on epidemiological models [The Lancet Infectious Diseases 21, no. 6 (2021): 793-802.].”

Reviewer 2 Report

It is very interesting that the authors showed that mathematical modelling has been suggested as an approach to optimising vaccine dose. This is a well-written paper, however, some points need clarifying and certain expression require further revised, there are given below,

1. In the introduction part, the background about mathematical modelling should be added in detail.

2. In the main text, some numbers such as “1.” (line 87) and “2.” (line 89) should be changed to “1)” and “2)”.

Author Response

It is very interesting that the authors showed that mathematical modelling has been suggested as an approach to optimising vaccine dose. This is a well-written paper, however, some points need clarifying and certain expression require further revised, there are given below,

We thank the reviewer for their time and expertise in reviewing this work.

  1. In the introduction part, the background about mathematical modelling should be added in detail.

We have added text to specify that we are discussing mathematical models, that these are equations/systems of equations, and that mathematical modelling based methodologies are well founded in literature from drug development. We agree with the reviewer that this may provide useful context for individuals without a mathematical modelling background.

  1. In the main text, some numbers such as “1.” (line 87) and “2.” (line 89) should be changed to “1)” and “2)”.

We thank the reviewer for this comment. In the manuscript that was uploaded we had used the formatting “1)”, however we believe that this has been updated to  the formatting “1.” by the journal. We have left numbered lists in the format “1.” as this is the formatting requested in the MDPI vaccine template.
